# Antimicrobial Lavandulylated Flavonoids from a Sponge-Derived *Streptomyces* sp. G248 in East Vietnam Sea

**DOI:** 10.3390/md17090529

**Published:** 2019-09-10

**Authors:** Duc Danh Cao, Thi Thanh Van Trinh, Huong Doan Thi Mai, Van Nam Vu, Hong Minh Le, Quyen Vu Thi, Mai Anh Nguyen, Thu Trang Duong, Dang Thach Tran, Van Minh Chau, Rui Ma, Gauri Shetye, Sanghyun Cho, Brian T. Murphy, Van Cuong Pham

**Affiliations:** 1Advanced Center for Bioorganic Chemistry, Institute of Marine Biochemistry, Vietnam Academy of Science and Technology (VAST), Hanoi 11307, Vietnam; saola43@gmail.com (D.D.C.); trtvan76@yahoo.com (T.T.V.T.); huongdm@imbc.vast.vn (H.D.T.M.); vuvannam87@gmail.com (V.N.V.); lhminhbk@gmail.com (H.M.L.); svuquyen@yahoo.com (Q.V.T.); maianhhsb@gmail.com (M.A.N.); duongtrang2901@gmail.com (T.T.D.); cvminh@vast.vn (V.M.C.); 2Faculty of Chemistry, Graduate University of Science and Technology, VAST, Hanoi 11307, Vietnam; 3Institute of Applied Science and Technology, University Industry of Vinh, Vinh 43115, Vietnam; trandangthach@yahoo.com.vn; 4Institute for Tuberculosis Research, College of Pharmacy, University of Illinois at Chicago, Chicago, IL 60612, USA; rma20@uic.edu (R.M.); gauris@uic.edu (G.S.); jkcno1@uic.edu (S.C.); 5Department of Pharmaceutical Sciences, College of Pharmacy, University of Illinois at Chicago, Chicago, IL 60607, USA; btmurphy@uic.edu

**Keywords:** actinomycete, *Streptomyces*, flavonoid, anti-tuberculosis, anti-microbial, cytotoxicity

## Abstract

Three new lavandulylated flavonoids, (2*S*,2″*S*)-6-lavandulyl-7,4′-dimethoxy-5,2′-dihydroxylflavanone (**1**), (2*S*,2″*S*)-6-lavandulyl-5,7,2′,4′-tetrahydroxylflavanone (**2**), and (2″*S*)-5′-lavandulyl-2′-methoxy-2,4,4′,6′-tetrahydroxylchalcone (**3**), along with seven known compounds **4–10** were isolated from culture broth of *Streptomyces* sp. G248. Their structures were established by spectroscopic data analysis, including 1D and 2D nuclear magnetic resonance (NMR), and high-resolution electrospray ionization mass spectrometry (HR-ESI-MS). The absolute configurations of **1–3** were resolved by comparison of their experimental and calculated electronic circular dichroism spectra. Compounds **1–3** exhibited remarkable antimicrobial activity. Whereas, two known compounds **4** and **5** exhibited inhibitory activity against *Mycobacterium tuberculosis* H37Rv with minimum inhibitory concentration (MIC) values of 6.0 µg/mL and 11.1 µg/mL, respectively.

## 1. Introduction

Marine sponges are known to harbor diverse microbial communities [1,2,3,4]. These marine-derived microorganisms have been a rich source for a multitude of biomedically relevant secondary metabolites [5,6,7]. Among actinomycetes, the genus *Streptomyces* is considered to be the most prolific producer of secondary metabolites for medical, agricultural and veterinary usage [8,9,10]. Previous reports showed that the *Streptomyces* genus has the ability to produce a variety of secondary metabolites with various activities, including antimicrobial, anticancer, and antiparasitic [11,12,13].

In search of bioactive metabolites from marine-derived actinomycetes, the *Streptomyces* sp. G248 (Appendix A), from the sponge *Halichondria panicea* (Pallas, 1766) collected from the coast of Da Nang in Vietnam was selected as its extract displayed antimicrobial activity against pathogenic *Enterococcus faecalis* and *Staphylococcus aureus* strains. Herein, we report the isolation and structural characterization of three new lavanludylated flavonoids (**1**–**3**) and seven known compounds (**4**–**10**) from the fermentation broth of the *Streptomyces* sp. G248 (Figure 1). Furthermore, their antimicrobial and cytotoxic activity evaluations were also discussed.

## 2. Results and Discussion

### 2.1. Isolation and Structural Elucidation

Culture broth solution (50 L) of *Streptomyces* sp. G248 was desalted by an Amberlite XAD-16 column, eluting with distilled water, then with methanol. The methanol solution was concentrated under reduced pressure. The crude extract (12.6 g) was purified by repeated open column chromatography to give compounds **1**–**10**.

Compound **1** was isolated as yellow amorphous solid, and was optically active {[α]^25^_D_ -5 (c 0.176, MeOH)}. Its positive HR-ESI mass spectrum (Appendix A) exhibited the proton adduct ion [M + H]^+^ at *m*/*z* 453.2270 (calcd. for C_27_H_33_O_6_, 453.2277) which, together with ^13^C NMR data, is consistent with the molecular formula of C_27_H_32_O_6_. The IR spectrum (Appendix A) indicated the absorption bands at ν_max_ 3366 cm^−1^ (OH) and 1696 cm^−1^, (C=O) functionalities. In the ^1^H NMR spectrum (Appendix A), the presence of an ABX system at δ_H_ 6.49 (d, *J* = 2.3 Hz, H-3′), 6.47 (dd, *J* = 2.3, 8.5 Hz, H-5′) and 7.38 (d, *J* = 8.5 Hz, H-6′), and a singlet proton at δ_H_ 6.12 (s, H-8) was observed at the aromatic region. Additionally, the resonances of protons at δ_H_ 5.55 (H-2), 4.96 (H-4″), 4.52 (H_a_-9″) and 4.59 (H_b_-9″), two methoxy groups at δ_H_ 3.82 and 3.83, three singlet methyls at δ_H_ 1.48 (CH_3_-6″), 1.57 (CH_3_-7″) and 1.64 (CH_3_-10″) and seven aliphatic protons from 1.80 to 3.10 ppm were also noted. Analysis of the ^13^C NMR spectrum (Appendix A) with the aid of an HSQC experiment (Appendix A) revealed 27 carbon resonances for **1**, including one ketone group (δ_C_ 193.6, C-4), ten sp^2^ quaternary carbons, five sp^2^ methines, one sp^2^ methylene, two sp^3^ methines, three sp^3^ methylenes, two methoxy groups and three methyls. Beside the aromatic ABX system, two other spin-spin coupling systems were revealed from the COSY spectrum (Appendix A): H-2 (δ_H_ 5.55)/CH_2_-3 (δ_H_ 2.67 and 2.89) (I), and CH_2_-1″ (δ_H_ 2.63)/H-2″ (δ_H_ 2.50)/CH_2_-3″ (δ_H_ 2.02)/H-4″ (δ_H_ 4.96) (II). The chemical shifts of carbons at δ_C_ 75.0 (C-2), 160.1 (C-2′), 159.0 (C-4′), 164.7 (C-5), 165.4 (C-7) and 161.9 (C-9) suggested their linkage to oxygen (Table 1). In the HMBC spectrum (Appendix A), cross-peaks of C-2″ (δ_C_ 48.2) with protons of CH_3_-10″ (δ_H_ 1.64) and CH_2_-9″ (δ_H_ 4.52 and 4.59), and those of C-8″ (δ_C_ 149.8) with protons of CH_3_-10″ indicated the linkage of the substructure II with the isopropenyl group through the C-2″/C-8″ bond. Additionally, HMBC correlations of C-4″ (δ_C_ 124.8) and C-5″ (δ_C_ 132.0) with protons of CH_3_-6″ (δ_H_ 1.48) and CH_3_-7″ (δ_H_ 1.57) defined the presence of a lavandulyl group in the structure of **1**. The presence of the A-ring was confirmed by the HMBC cross-peaks of H-8 (δ_H_ 6.12) with C-6 (δ_C_ 109.6), C-7 (δ_C_ 165.4), C-9 (δ_C_ 161.9) and C-10 (δ_C_ 105.7). The aromatic ABX system (B-ring) was confirmed by HMBC correlations of H-3′ (δ_H_ 6.49) with C-1′ (δ_C_ 119.7), C-2′ (δ_C_ 160.1), C-4′ (δ_C_ 159.0) and C-5′ (δ_C_ 108.1). The connection of C-2 with the B-ring at C-1′ was revealed by cross-peaks of H-2 (δ_H_ 5.55) with C-1′ (δ_C_ 119.7) and C-6′ (δ_C_ 128.5). The spectral features of coupling systems I and the carbonyl group deduced from HMBC signals indicated structural similarity of **1** to flavanone compounds. Furthermore, the attachment of the lavandulyl group to the A-ring at C-6 via C-6/C-1″ linkage was revealed by HMBC cross-peaks of the protons of CH_2_-1″ (δ_H_ 2.63) with C-5 (δ_C_ 164.7), C-6 (δ_C_ 109.6) and C-7 (δ_C_ 165.4), and was supported by the correlations of the methoxy protons at C-7 with the protons of CH_2_-1″) in the ROESY spectrum of **1** (Appendix A). Finally, the two methoxy groups at δ_H_ 3.82 and 3.83 were attached to C-7 and C-4′ as indicated by their correlations in the HMBC spectrum (Figure 2). Complete analyses of 2D NMR spectra established the planar structure of **1** as 6-lavandulyl-7,4′-dimethoxy-5,2′-dihydroxylflavanone.

The 1D NMR spectra of compound **2** (Appendix A) displayed resonances close to those of **1**, except the absence of two methoxy groups. Complete analysis of 2D NMR spectra (Appendix A) established the planar structure of **2** as 6-lavandulyl-5,7,2′,4′-tetrahydroxylflavanone (Appendix A), which shared the same planar structure with kushnol F, previously isolated from *Sophora flavescens* [14]. However, the absolute configuration of the chiral carbon C-2″ has not been determined. Moreover, compound **2** had an optical rotation activity of +3 (*c* 0.2, MeOH), while this value is -59.8 (*c* 0.82, MeOH) for kushnol F [14]. 

Compound **3** was isolated as yellow amorphous solid and optically active {[α]^25^_D_ -1.8 (*c* 0.54, MeOH}. Its positive HR-ESI-MS (Appendix A) showed the proton adduct ion [M + H]^+^ at *m*/*z* 439.2117 (calcd. for C_26_H_31_O_6_, 439.2121), consistent with a molecular formula of C_26_H_30_O_6_. The 1D NMR spectra of **3** (Appendix A) exhibited the resonances for the A- and B-ring systems, and the lavandulyl group as found in compounds **1** and **2**. However, signals of a CH=CH system were observed in the 1D NMR spectra of **3** instead of resonances of the CH-2/CH_2_-3 spin system in the structures of **1** and **2**. 

Analysis of 2D NMR spectra of **3** (Appendix A) confirmed the presence of the A and B-rings, and the lavandulyl substructure. Additionally, cross-peaks of H-β at δ_H_ 8.02 with C-2 (δ_C_ 160.3), C-6 (δ_C_ 131.6) and carbonyl carbon C-γ (δ_C_ 194.8) in the HMBC spectrum (Appendix A) indicated a chalcone skeleton for **3** (Appendix A). The methoxy group at C-2′ was revealed by an HMBC correlation of C-2′ at δ_C_ 162.3 with the methoxy protons at δ_H_ 3.91 (Figure 3). As in the case of compounds **1** and **2**, the location of the lavandulyl group at C-5′ was established by the HMBC cross-peaks of C-4′ (δ_C_ 164.1), C-5′ (δ_C_ 108.9) and C-6′ (δ_C_ 166.6) with the protons of CH_2_-1″ (δ_H_ 2.65). The planar structure of **3** was finalized as 5′-lavandulyl-2′-methoxy-2,4,4′,6′-tetrahydroxylchalcone. This compound has the same planar structure as kuraridin, which was previously isolated from *Albizzia julibrissin* [15]. The optical rotation activity of kuraridin {[α]_D_^20^ +1.2 (*c* 0.025, MeOH)} [13] is opposite to that of compound **3** {[α]_D_^25^ –1.8 (*c* 0.54, MeOH)}, and the absolute configuration at C-2″ of kuraridin has not been reported.

Absolute configurations of the compounds **1**–**3** were resolved by comparison of their experimental and calculated electronic circular dichroism (ECD) spectra [16,17]. The ECD quantum chemical calculations were performed using the Gaussian 09 software [18]. To obtain minimum energy conformers, geometry optimization of each possible isomer of these compounds was conducted (Appendix A). The calculated ECD spectra of compounds **1**–**3** were generated using the time-dependent density functional method at the B3LYP/LanL2DZ level. Since the circular dichroism (CD) spectrum of compound **1** (Appendix A) had a positive Cotton effect at 337 nm (Δε +3.6) and a negative Cotton effect at 291 nm (Δε −7.7), the *S*-configuration was thus assigned for carbon C-2 of **1** [19]. The ECD spectra of the two possible diastereomers (2*S*,2″*S*)-**1** and (2*S*,2″*R*)-**1** were generated. As shown in Figure 4, the ECD spectrum of (2*S*,2″*S*)-**1** was similar as the experimental CD spectrum of **1**, while an ECD spectrum of (2*S*,2″*R*)-**1** displayed the opposite Cotton effect at 233 nm. The *S*-configuration was thus assigned for the chiral carbon C-2″ of **1**. Similarly, the *S*-configuration at C-2 was determined for **2** by the observation of a positive Cotton effect at 315 nm (Δε +0.8) and a negative Cotton effect at 291 nm (Δε −5.9) in the CD spectrum of **2** (Appendix A). By comparison of the ECD spectra of the two diastereomers (2*S*,2″*S*)-**2** and (2*S*,2″*R*)-**2** with the experimental CD spectrum of **2** (Figure 5), the *S*-configuration was also assigned for C-2″ of **2**. The ECD of (2″*S*)-**3** was in good agreement with the experimental CD spectrum of **3** (Appendix A). This observation allowed determination the *S*-configuration for carbon C-2″ for **3** (Figure 6). Since the optical rotation activities of **2** and **3** are comparatively opposite with kushnol F and kuraridin, respectively, the absolute configuration at C-2″ of kushnol F and kuraridin would most likely be *R*, however further experiments are required to state this definitively.

Other known compounds, (2S,2″S)-6-lavandulyl-7-methoxy-5,2′,4′-trihydroxylflavanone (**4**) [20], 6-prenyl-4′-methoxy-5,7-dihydroxylflavanone (**5**) [21], norharman (β-carboline) (**6**) [22,23], cyclo-(Pro-Leu) (**7**) [24], cyclo-(Pro-Phe) (**8**) [25], cyclo-(Pro-Tyr) (**9**) [26], and adenine (**10**) [27] were also isolated and characterized from the culture broth of G248 strain. Their structures were established by spectroscopic data analysis and comparison with data reported in the literature.

### 2.2. Biological Activity

Compounds **1**–**10** were evaluated for antibacterial activity against *Enterococcus faecalis* (ATCC13124), *Staphylococcus aureus* (ATCC25923), *Bacillus cereus* (ATCC13245), *Escherichia coli* (ATCC25922), *Pseudomonas aeruginosa* (ATCC27853), *Salmonella enterica* (ATCC12228), antiyeast activity against *Candida albicans* (ATCC1023), and in a separate experiment for anti-TB activity against *Mycobacterium tuberculosis* H37Rv. Compounds **1**–**3** exhibited antimicrobial activity, **2** exhibiting the most potent activity profile of the group. Compound **3** seems to have a broad spectrum of antimicrobial activity as it was active against all tested microbial strains (with the exception of *M. tuberculosis* H37Rv). Except against *E. coli*, compounds **1** and **2** showed significant inhibition activity against the remaining tested strains. The two known compounds **4** and **5** exhibited IC_90_ values of 6.0 and 11.1 µg/mL against *M. tuberculosis* H37Rv, respectively. This is the first report of the anti-TB activity for these two compounds (Table 2). Since, compound **4** exhibited anti-TB activity while **1** and **2** had no effect against *M. tuberculosis*, the methoxy group at C-7 and hydroxyl functionality at C-4′ seem to be critical for the anti-TB activity of these compounds. Cytotoxicity evaluation indicated that compounds **4** and **5** had inhibition against four tested cancer cell lines, KB, Hep-G2, Lu-1 and MCF-7 with IC_50_ values from 2.0 to 14.6 µg/mL (Table 3). Other compounds had weak or no cytotoxicity against these four tested cancer cell lines.

## 3. Materials and Methods

### 3.1. General Experimental Procedures

Optical rotations were recorded on a Polax-2L polarimeter in MeOH. HR-ESIMS were recorded on a FT-ICR 910-MS TQFTMS-7 T mass spectrometer. CD spectra were taken on a Chirascan CD spectrometer. IR spectra were recorded on a Nicolet Impact 410 FT-IR spectrometer, and NMR spectra on a Bruker AM500 MHz spectrometer operating at 125.76 MHz for ^13^C NMR, and at 500.13 MHz for ^1^H NMR. ^1^H chemical shifts were referenced to CD_3_OD at δ 3.31 ppm, while the ^13^C chemical shifts were referenced to the central peak at δ 49 ppm. For HMBC experiments the delay (1/2J) was 70 ms.

### 3.2. Actinomycete Material 

The actinomycete strain G248 was obtained from the *Halichondria panicea* (Pallas, 1766) sponge (sponge taxonomy was identified by Prof. Do Cong Thung, Institute of Marine Environment and Resources—Vietnam Academy of Science and Technology), collected in Son-Tra island (Da Nang)—Vietnam in August 2016. A voucher specimen was deposited at the Institute of Marine Environment and Resources, Vietnam Academy of Science and Technology (VAST).

The strain G248 was isolated from above-mentioned sponge sample. The taxonomy of the strain G248 was identified by using 16S rRNA gene sequence analysis. Its gene sequence was registered with GenBank access code MG917690 (Appendix A). On the basis of morphological and phylogenetic evidence (Appendix A), the actinomycete strain G248 was assigned to the genus *Streptomyces*.

### 3.3. Fermentation and Extraction

Strain G248 was activated and inoculated into 1 L of A1 medium pH 7.0 comprising starch (5.0 g), yeast extract (2.0 g), peptone (1.0 g) and artificial sea salt (30.0 g) in 1.0 L of distilled water. After 7 days of incubation at 28 °C with agitation, the culture broth was used to inoculate the fermentation in 50 L of high-nutrient medium A1^+^ (soluble starch: 10 g/L; yeast extract: 4 g/L; peptone: 2 g/L; instant ocean: 30 g/L; CaCO_3_: 1 g/L; agar: 15 g/L; add water to 1 L, pH 7.0). The fermentation was incubated at 28 °C with agitation of 200 rpm and harvested on the tenth day.

### 3.4. Isolation and Purification

The fermentation broth (50 L) was passed through a XAD-16 column (10 kg XAD-16). The column was washed with distilled water (70 L), followed by eluting with methanol (80 L). The methanol solution was concentrated under reduced pressure. The crude extract (12.6 g) was purified by column chromatography (CC) on silica gel, eluted with CH_2_Cl_2_/EtOH gradient to give 11 fractions. Fraction F4 (1.23 g) was separated by CC on Sephadex LH-20 (CH_2_Cl_2_/MeOH: 9/1), providing 4 subfractions. Subfraction F4.3 (90 mg) was purified by CC on silica gel (0 to 50% MeOH in CH_2_Cl_2_) to furnish compounds **7** (5 mg). Fraction F5 (1.72 g) was subjected to CC on silica gel (0 to 100% MeOH in CH_2_Cl_2_), leading to four subfractions. Subfraction F5.3 (305 mg) was separated by CC on silica gel, eluted with a CH_2_Cl_2_/acetone gradient, providing 4 sub-subfractions. Sub-subfraction F5.3.4 was chromatographed on silica gel, eluted with EtOAc/MeOH gradient, giving compounds **6** (5 mg) and **8** (15 mg). Fraction F6 (550 mg) was purified by CC on silica gel (0 to 100% MeOH in CH_2_Cl_2_) providing 4 subfractions. Subfraction F6.4 (1.16 g) was separated by CC on Sephadex LH-20 (MeOH/CH_2_Cl_2_: 9/1), followed by preparative thin-layer chromatography (TLC) (CH_2_Cl_2_/MeOH: 8.5/1.5) to furnish compounds **9** (3.9 mg). Fraction 9 (360 mg) was separated by CC on Sephadex LH-20 (MeOH/CH_2_Cl_2_: 9/1), leading to 7 subfractions. Subfraction F9.5 (35 mg) was purified by CC on silica gel (0 to 100% MeOH in EtOAc), followed by preparative TLC (EtOAc/MeOH: 8/2) to furnish compound **10** (5.9 mg). Fraction 10 (1.62 g) was separated by CC on silica gel (0 to 100% MeOH in CH_2_Cl_2_), furnishing six subfractions. Subfraction F10.2 (55 mg) was subjected to CC on Sephadex LH-20 (MeOH/CH_2_Cl_2_: 9/1), followed by preparative TLC (CH_2_Cl_2_/acetone: 7/3) to yield compounds **1** (2.5 mg) and **5** (3.7 mg). Subfraction F10.4 (40 mg) was chromatographed on Sephadex LH-20 column (MeOH/CH_2_Cl_2_: 9/1), affording compounds **2** (3.0 mg) and **3** (6.0 mg). Finally, subfraction F10.5 (70 mg) was subjected to CC on Sephadex LH-20 (MeOH/CH_2_Cl_2_: 9/1), followed by preparative TLC (CH_2_Cl_2_/acetone: 7/3) to afford compound **4** (3.9 mg).

### 3.5. Spectral Data 

*(2S,2″S)-6-lavandulyl-7,4′-dimethoxy-5,2′-dihydroxylflavanone* (**1**)—amorphous yellow solid; [α]_D_^25^ -5 (*c* 0.176, MeOH); R_f_: 0.7 (CH_2_Cl_2_-MeOH, 8.5:1.5); IR (KBr): 3366, 2916, 1696, 1497, 1457, 1411, 1376, 1278, 1151, 1093, 974, 886 cm^−1^; ^1^H NMR (CD_3_OD, 500 MHz) and ^13^C NMR (CD_3_OD, 125 MHz), Table 1; CD (MeOH) nm (Δε): 205 (−10.2), 225 (+12.8), 259 (+0.7), 291 (−7.7), 337 (+3.6); HR-ESI-MS *m*/*z* 453.2270 [M + H]^+^ (calcd. for C_27_H_33_O_6_, 453.2277).

(2S,2″S)-6-lavandulyl-5,7,2′,4′-tetrahydroxylflavanone (**2**)—amorphous yellow solid; [α]_D_^25^ +3 (c 0.2, MeOH); R_f_: 0.45 (CH_2_Cl_2_-MeOH, 8.5:1.5); IR (KBr): 3261, 2961, 2921, 1631, 1604, 1514, 1439, 1382, 1296, 1162, 1083, 974, 887 cm^−1^; ^1^H NMR (CD_3_OD, 500 MHz) and ^13^C NMR (CD_3_OD, 125 MHz), Table 1; CD (MeOH) nm (Δε): 204 (−6.5), 220 (+8.4), 258 (+0.6), 291 (−5.9), 315 (+0.8); HR-ESI-MS m/z 425.1960 [M + H]^+^ (calcd. for C_25_H_29_O_6_, 425.1964). 

(*2″S)-5′-lavandulyl-2′-methoxy-2,4,4′,6′-tetrahydroxylchalcone* (**3**)—amorphous yellow solid; [α]_D_^25^ -1.8 (*c* 0.54, MeOH); R_f_: 0.33 (CH_2_Cl_2_-MeOH, 8.5:1.5); IR (KBr): 3695, 2925, 2854, 1605, 1546, 1449, 1233, 1141, 1104 cm^−1^; ^1^H NMR (CD_3_OD, 500 MHz) and ^13^C NMR (CD_3_OD, 125 MHz), Table 1; CD (MeOH) nm (Δε): 212 (+0.9), 259 (+0.5), 291 (−0.2), 319 (+0.06); HR-ESI-MS *m*/*z* 439.2117 [M + H]^+^ (calcd. for C_26_H_31_O_6_, 439.2121).

### 3.6. Antimicrobial Activity Assay

Antimicrobial assays were carried out using *E. coli* (ATCC25922), *P. aeruginosa* (ATCC27853), *S. enterica* (ATCC12228), *E. faecalis* (ATCC13124), *S. aureus* (ATCC25923), *B. cereus* (ATCC13245), and *C. albicans* (ATCC1023). Stock solutions of samples were prepared in DMSO, and the antimicrobial assays were carried out in 96-well microtiter plates against the microbial strains (5 × 10^5^ CFU/mL) using a modification of the published method [28]. After incubation for 24 h at 37 °C, the absorbance at 650 nm was measured using a microplate reader. Streptomycin and nystatin were used as reference compounds.

### 3.7. Anti-Mycobacterial Activity Assay

*M. tuberculosis* H37Rv (ATCC 27294) was purchased from American Type Culture Collection (ATCC). These strains were cultured to late log phase in Middlebrook 7H9 broth supplemented with 0.2% (vol/vol) glycerol, 0.05% Tween™ 80, and 10% (*v*/*v*) oleic acid-albumin-dextrose-catalase (OADC). The culture was harvested and resuspended in phosphate-buffered saline. Suspensions were then filtered through 8 μm filter membranes and frozen at ‒80 °C. Prior to use of bacterial stocks for the anti -TB assay, colony-forming units (CFUs) were determined by plating on 7H11 agar media. The MIC is defined here as the lowest concentration resulting in ≥90% growth inhibition of the bacteria relative to untreated controls. MIC against replicating *M. tuberculosis* was measured by the Microplate Alamar Blue Assay (MABA) in 7H12 media [29].

### 3.8. Cytotocicity Assay

An MTT assay was used to determine the cytotoxic activity of compounds **1**–**10** with human cancer cell lines (KB, LU-1, Hep-G2 and MCF-7) acquired from the American Type Culture Collection (ATCC, Manassas, VA) using a modification of the published method [30]. Cells were cultured in medium RPMI 1640 supplemented with 10% FBS (fetal bovine serum) under a humidified atmosphere of 5% CO_2_ at 37 °C. Compounds **1**–**10** were dissolved in DMSO at a concentration of 20 mg/mL. A series of dilutions for each compound was prepared to final concentrations of 128, 32, 8, 2 and 0.5 mg/mL. Samples (100 µL) of the complexes with different concentrations were added to the wells on 96-well plates. Cells were separated with trypsin and ethylenediaminetetraacetic acid (EDTA), and seeded in each well with 3 × 10^4^ cells per well. An MTT solution (20 µL, 4 mg/mL) of phosphate buffer saline (8 g NaCl, 0.2 g KCl, 1.44 g Na_2_HPO_4_ and 0.24 g KH_2_PO_4_ per L) was added to each well after being incubated for 48 h. The cells were further incubated for 4 h and a purple formazan precipitate was formed, which was separated by centrifugation. The precipitate was dissolved by adding DMSO (100 µL) to each well. The optical density of the solution was determined by a plate reader (TECAN) at 540 nm. The inhibition ratio was achieved on the basis of the optical densities from the calculation of three replicate tests.

## Figures and Tables

**Figure 1 marinedrugs-17-00529-f001:**
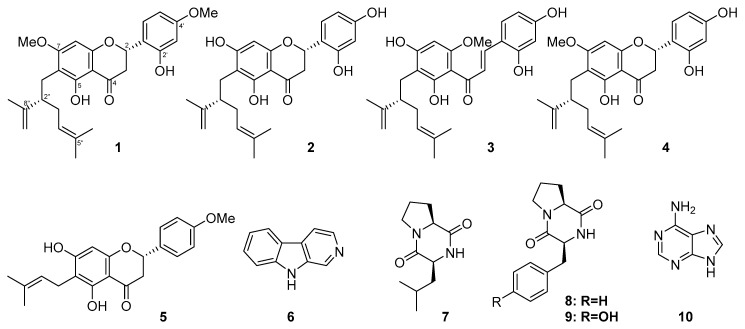
Structures of compounds **1**–**10**.

**Figure 2 marinedrugs-17-00529-f002:**
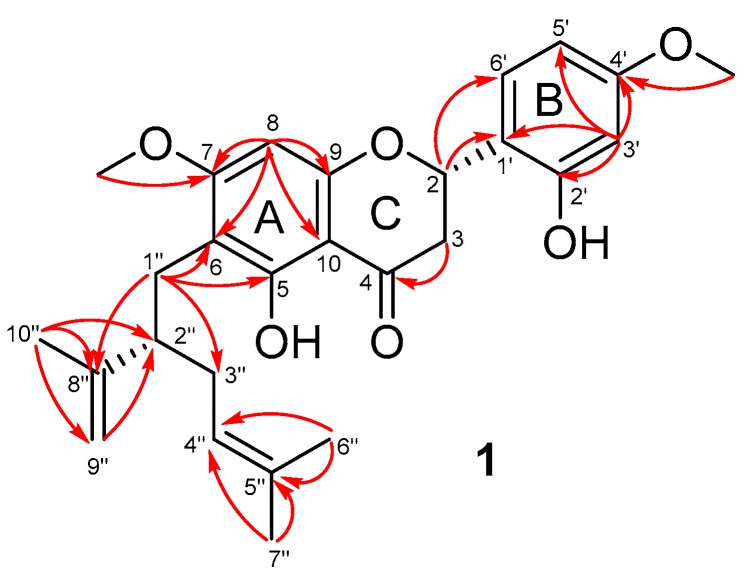
Key HMBC correlations of **1**.

**Figure 3 marinedrugs-17-00529-f003:**
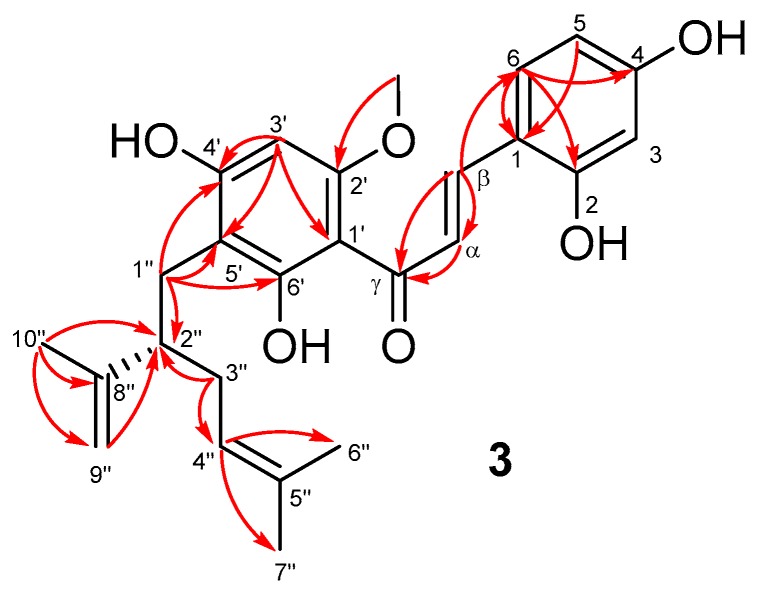
Selected HMBC correlations of **3**.

**Figure 4 marinedrugs-17-00529-f004:**
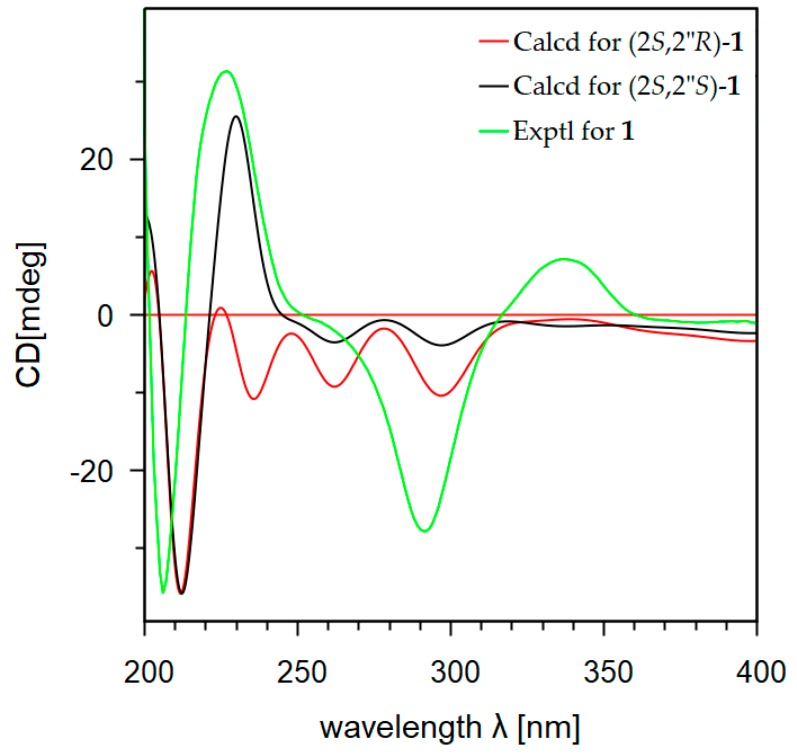
Calculated ECD and experimental CD spectra of **1**.

**Figure 5 marinedrugs-17-00529-f005:**
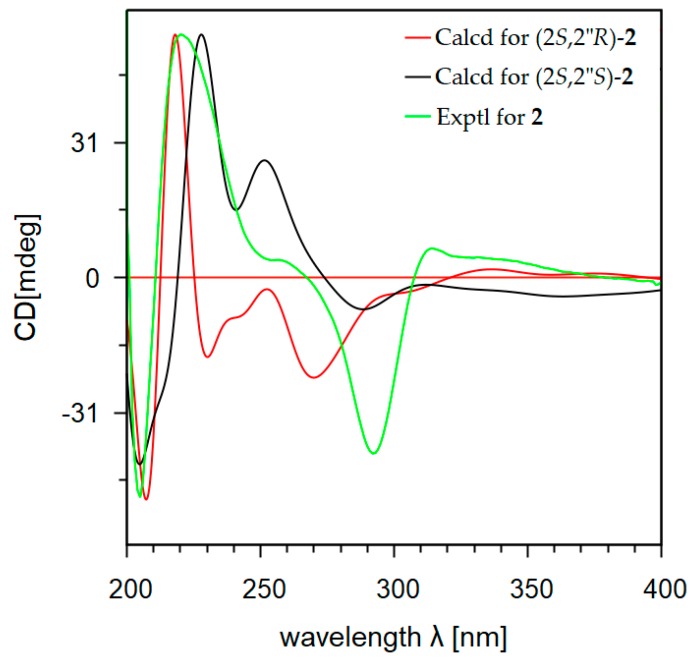
Calculated ECD and experimental CD spectra of **2**.

**Figure 6 marinedrugs-17-00529-f006:**
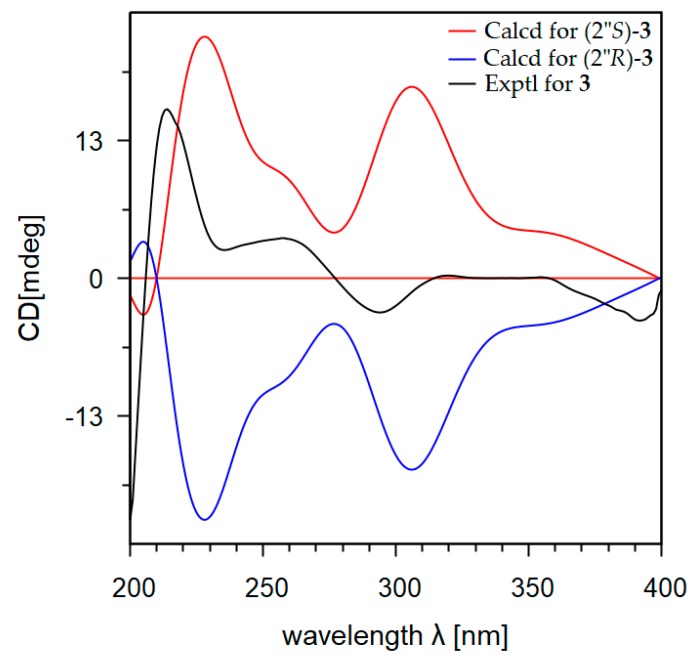
Calculated ECD and experimental CD spectra of **3**.

**Table 1 marinedrugs-17-00529-t001:** NMR data for compounds **1**–**3** (CD_3_OD, ^1^H: 500 MHz, ^13^C: 125 MHz).

	1		2		3	
N	δ_C_	δ_H_ mult. (*J* in Hz)	δ_C_	δ_H_ mult. (*J* in Hz)	δ_C_	δ_H_ mult. (*J* in Hz)
1	-	-	-	-	116.3	-
2	75.0	5.55 dd (2.5, 13.5)	75.8	5.58 dd (3.0, 13.0)	160.3	-
3	45.4	2.89 dd (13.5, 17.0) 2.67 dd (2.5, 17.0)	43.3	2.99 dd (13.0, 17.0) 2.75 dd (3.0, 17.0)	103.7	6.35 br.s
4	193.6	-	198.9	-	162.4	-
5	164.7	-	162.6	-	109.0	6.36 dd (2.0, 8.0)
6	109.6	-	108.7	-	131.6	7.41 d (8.0)
7	165.4	-	167.0	-	-	-
8	93.6	6.12 s	96.4	5.93 s	-	-
9	161.9	-	163.2	-	-	-
10	105.7	-	103.2	-	-	-
1′	119.7	-	118.4	-	106.5	-
2′	160.1	-	156.7	-	162.3	-
3′	99.8	6.49 d (2.3)	103.4	6.36 d (2.5)	91.6	6.02 s
4′	159.0	-	159.6	-	164.1	-
5′	108.1	6.47 dd (2.3, 8.5)	107.7	6.37 dd (2.0, 8.5)	108.9	-
6′	128.5	7.38 d (8.5)	128.6	7.32 d (8.5)	166.6	-
1″	28.2	2.63 m	28.0	2.60 m	28.2	2.65 m
2″	48.2	2.50 m	48.4	2.49 m	48.0	2.57 m
3″	32.3	2.02 m	32.3	2.01 m	32.4	2.09 m
4″	124.8	4.96 t (5.5)	124.8	4.99 t (5.5)	125.0	5.06 td (1.0, 6.5)
5″	132.0	-	132.0	-	131.8	-
6″	17.8	1.48 s	17.8	1.50 s	17.9	1.58 s
7″	25.8	1.57 s	25.8	1.59 s	25.9	1.65 s
8″	149.8	-	149.8	-	149.9	-
9″	111.2	4.52 s 4.59 s	111.1	4.54 br s 4.60 br s	111.1	4.61 br s 4.55 br s
10″	19.1	1.64 s	19.2	1.65 s	19.1	1.72 s
OMe	55.9	3.82 s	-	-	-	-
OMe	55.9	3.83 s	-	-	56.1	3.91
α	-	-	-	-	125.4	7.95 d (16.0)
β	-	-	-	-	139.8	8.02 d (16.0)
γ	-	-	-	-	194.8	-

**Table 2 marinedrugs-17-00529-t002:** Antimicrobial activities of compounds **1**–**10** (IC_90_: μg/mL).

Compd.	*E. faecalis*	*S. aureus*	*B. cereus*	*E. coli*	*P. aeruginosa*	*S. enterica*	*C. albicans*	*M. tuberculosis*
**1**	8	8	8	>256	8	8	16	48.0
**2**	1	1	1	>256	1	8	1	>50
**3**	8	8	8	4	8	8	16	>50
**4**	32	32	16	128	32	32	32	6.0
**5**	>256	>256	>256	>256	>256	>256	>256	11.1
**6**	>256	>256	>256	128	>256	>256	>256	>50
**7**	>256	>256	>256	>256	>256	>256	>256	>50
**8**	>256	>256	>256	>256	>256	>256	>256	>50
**9**	>256	>256	>256	>256	>256	>256	>256	>50
**10**	32	nt	nt	128	nt	nt	64	>50
Strep	256	256	128	32	256	128	nt	nt
Cyclohex	nt	nt	nt	nt	nt	nt	32	nt

Strep: Streptomycin; Cyclohex: Cyclohexamide; nt: not tested.

**Table 3 marinedrugs-17-00529-t003:** Cytotoxic activity of compounds **1**–**10** (IC_50_: μg/mL).

Compd	KB	Hep-G2	Lu-1	MCF-7
**1**	59.7	32.0	80.0	71.7
**2**	118.4	>128	>128	>128
**3**	>128	>128	>128	>128
**4**	4.8	2.27	4.0	14.5
**5**	2.0	2.0	4.8	11.8
**6**	47.3	78.4	71.5	60.0
**7**	>128	56.0	>128	>128
**8**	>128	>128	>128	>128
**9**	>128	>128	>128	>128
Ellipticine	0.3 ± 0.05	0.3 ± 0.05	0.4 ± 0.05	0.5 ± 0.05

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
