# Peer review of "Antimicrobial Lavandulylated Flavonoids from a Sponge-Derived Streptomyces sp. G248 in East Vietnam Sea"

_marinedrugs, 2019, doi:10.3390/md17090529_

Round 1

Reviewer 1 Report

Sponge associated microbes particularly actinomycete is very promising source as a bioactive compound. If the original source of compounds come from microbes then there is scope to scale up. The author did nice work by elucidating new bioactive compounds from Streptomyces sp. G248. Compound 3 looks very promising as an antibacterial agent against gram negative bacteria.

Minor:

Line 28: Write down the elaboration of TB

Line 31: Please incorporate more appropriate references for this statement.

Line 246: 5x105 please make correction with this.

Author Response

Line 28: Anti-TB was corrected by Anti-tuberculosis

Line 32: Two more references were added.

line 246: 5x105 was correctly corrected  

Reviewer 2 Report

Please re-organize the 2nd paragraph of "Introduction", as "Herein we report the isolation and structural elucidation of *** from the fermentation broth of the streptomyces sp. G248" appeared twice. The optical rotation of all 3 new compounds are of small number comparing with the known molecules, which were used for comparsion. Have you tried to analyze these 3 new compounds with HPLC equipped with chiral columns? Just to rule out the possibilities of being mixture of enantiomers.

Author Response

The second paragraph of the Introduction section was revised.

We did not analyse these compounds by HPLC. However, we are sure that these three compounds 1-3 are not racemic mixtures as indicated by their CD spectra. CD spectrum of a racemic mixture has not Cotton effects.

Reviewer 3 Report

The paper “Antimicrobial Lavandulylated Flavonoids from a Sponge‐Derived Streptomyces sp. G248 in East Vietnam Sea” describes the structure characterization of three lavandulylated flavonoids by NMR and ECD to suggest the absolute stereochemistry.

I recognize that the document is well written, but in my opinion, there is no much novelty in the chemistry reported: compound 1 differing from compound 4, already reported by the same authors from a different strain, only for the lack of a methoxy group, whereas compounds 2 and 3 presumably differ from the known Kushnol F and Kuraridin respectively only for the stereochemistry od C-2”, relying only on the optical rotation. Even if theorically when the specific rotation of a pure chiral compound is known, it is possible to use the observed specific rotation to determine the "optical purity", the compounds have been isolated by open column or preparative TLC (no HPLC) and the presence of small amounts of highly rotating impurities can greatly affect the rotation of samples, thus in practice the efficacy of this method is limited. I would like you to notice that the [α]D20 of compound 1 and 2, that from ECD seem to possess the same stereochemistry of the two chiral centers, have opposite signal and value very close to the zero.

Moreover, the make the test clearer I would suggest inserting in Figure 1 at list few key numbers on structure 1, to help the reader to follow the description before to arrive to figure 2.

I would suggest the authors increase the value of the paper investigating the biological activity and the mechanism of action of anti-tuberculosis active compounds.

Author Response

The absolute configurations of both kushnol F and kuraridin have not been determined yet. Our two compounds 2 and 3 were established their absolute configurations by comparison of their experimental and calculated electronic circular dichroism spectra.

The optical rotation activities of compounds 1-3 are small. However they are not racemic mixtures. The CD spectra of 1-3 displayed strong Cotton effects that indicated they are not racemic compounds.

Some key nunbers of compound 1 were inserted in the Figure 1.

The study of anti-TB mechanism of action needs more quantities and time. This would be a further project. 

Round 2

Reviewer 3 Report

About the absolute configurations of kushnol F and kuraridin , in my opinion, authors should be more cautious saying that they can “suggest” it.

Regarding the optical rotation, I agree with authors that the strong Cotton effects suggest that the samples are not racemic mixtures, but anyway small amounts of different highly rotating impurities can greatly affect the rotation of a given sample and as is visible from the NMR spectra, all the samples contain very small signals that don’t belong to the compounds, and this is usual for samples that are not purified by HPLC. Eventually, authors might do an HPLC purification and repeat the optical rotation of the samples to confirm the data.